# ATP-dependent chromatin assembly is functionally distinct from chromatin remodeling

Sharon E Torigoe[1], Ashok Patel[2], Mai T Khuong[1], Gregory D Bowman[2], James T Kadonaga[1]*

[1]Section of Molecular Biology, University of California, San Diego, La Jolla, United States; [2]TC Jenkins Department of Biophysics, Johns Hopkins University, Baltimore, United States

**Abstract** Chromatin assembly involves the combined action of ATP-dependent motor proteins and histone chaperones. Because motor proteins in chromatin assembly also function as chromatin remodeling factors, we investigated the relationship between ATP-driven chromatin assembly and chromatin remodeling in the generation of periodic nucleosome arrays. We found that chromatin remodeling-defective Chd1 motor proteins are able to catalyze ATP-dependent chromatin assembly. The resulting nucleosomes are not, however, spaced in periodic arrays. Wild-type Chd1, but not chromatin remodeling-defective Chd1, can catalyze the conversion of randomly-distributed nucleosomes into periodic arrays. These results reveal a functional distinction between ATP-dependent nucleosome assembly and chromatin remodeling, and suggest a model for chromatin assembly in which randomly-distributed nucleosomes are formed by the nucleosome assembly function of Chd1, and then regularly-spaced nucleosome arrays are generated by the chromatin remodeling activity of Chd1. These findings uncover an unforeseen level of specificity in the role of motor proteins in chromatin assembly.

## Introduction

*For correspondence:
jkadonaga@ucsd.edu

The assembly of nucleosomes is necessary for the regeneration of chromatin following DNA replication, transcription, and DNA repair, and is an active ATP-driven process, as originally discovered by Worcel et al. (*Glikin et al., 1984*; *Ruberti and Worcel, 1986*). Nucleosome assembly is facilitated by the combined activities of ATP-dependent motor proteins (reviewed in *Haushalter and Kadonaga, 2003*; *Lusser and Kadonaga, 2004*) and histone chaperones (reviewed in *Corpet and Almouzni, 2009*; *Campos and Reinberg, 2010*; *Das et al., 2010*; *Ransom et al., 2010*; *Avvakumov et al., 2011*; *Elsässer and D'Arcy, 2012*; *Burgess and Zhang, 2013*). Histone chaperones initially deposit core histones onto DNA to form non-nucleosomal histone–DNA intermediates (prenucleosomes), which can then be converted into periodic arrays of canonical nucleosomes by ATP-driven motor proteins (*Torigoe et al., 2011*).

ATP-dependent factors that participate in chromatin assembly include Chd1 (chromo-ATPase/helicase-DNA-binding protein 1), ATRX (alpha thalassemia/mental retardation syndrome X-linked), and several ISWI (imitation switch)-containing complexes, such as ACF (ATP-utilizing chromatin assembly and remodeling factor), CHRAC (chromatin accessibility complex), RSF (remodeling and spacing factor), and ToRC (Toutatis-containing chromatin remodeling complex) (*Ito et al., 1997*; *Varga-Weisz et al., 1997*; *Loyola et al., 2001*; *Lusser et al., 2005*; *Lewis et al., 2010*; *Emelyanov et al., 2012*). These motor proteins exhibit both chromatin assembly and remodeling activities, and are members of the SNF2 (sucrose non-fermenting 2) protein family, which comprises the ATPases that are known to be involved in chromatin remodeling (reviewed in *Clapier and Cairns, 2009*; *Flaus and Owen-Hughes,*

**eLife digest** In many cells, genomic DNA is wrapped around proteins known as histones to produce particles called nucleosomes. These particles then join together—like beads on a string—to form a highly periodic structure called chromatin. In the nucleus, chromatin is further folded and condensed into chromosomes. However, many important processes, including the replication of DNA and the transcription of genes, require access to the DNA. The cell must therefore be able to disassemble chromatin and remove the histones, and then, once these processes are complete, to reassemble the chromatin. Enzymes known as chromatin assembly factors are responsible for the disassembly and reassembly of chromatin.

There are two main types of chromatin assembly factors in eukaryotic cells (i.e., cells with nuclei)—histone chaperones and motor proteins. The histone chaperones escort histones from the cytoplasm, where they are made, to the nucleus. The motor proteins—using energy supplied by ATP molecules—then catalyze the formation of nucleosomes. This involves two activities: the motor proteins assemble nucleosomes by helping the DNA to wrap around the histones, and they also remodel chromatin by altering the positions of nucleosomes along the DNA to ensure that they are periodic—that is, regularly spaced.

A conserved motor protein called Chd1 performs chromatin assembly and remodeling in eukaryotic cells. Chd1 works in conjunction with histone chaperones—both are needed for chromatin assembly, and so are DNA, histones and ATP. However, whether or not chromatin assembly and chromatin remodeling by Chd1 are identical or distinct processes is not well understood.

Torigoe et al. have now discovered a mutant Chd1 protein that has nucleosome assembly activity (i.e., it can make nucleosomes) but cannot remodel chromatin (i.e., it is unable to move nucleosomes), and thus have demonstrated that these two processes are functionally distinct. Torigoe et al. additionally have found that the mutant Chd1 proteins produce randomly distributed nucleosomes rather than the periodic arrays normally found in chromatin. Further analysis then revealed that the wild-type Chd1 protein, which can remodel chromatin, is able to convert randomly distributed nucleosomes into periodic arrays.

These findings have led to a new model for chromatin assembly in which Chd1 initially generates randomly distributed nucleosomes (via its assembly function), and then converts them into periodic arrays of nucleosomes (via its remodeling function). Together, these studies shed light on the mechanisms by which chromatin is created and manipulated in cells.

2011; *Hargreaves and Crabtree, 2011*; *Ryan and Owen-Hughes, 2011*). With the Chd1, ACF, and ToRC motor proteins, it has been shown that efficient chromatin assembly requires both a histone chaperone (such as NAP1) and the motor protein (*Ito et al., 1997*, *1999*; *Lusser et al., 2005*; *Emelyanov et al., 2012*). Hence, the Chd1, ACF, and ToRC motor proteins are not able to catalyze chromatin assembly in the absence of a histone chaperone. To determine whether nucleosome assembly requires the ability to reposition nucleosomes, we examined the properties of mutant Chd1 proteins that are defective for chromatin remodeling activity yet retain a substantial amount of their ATPase activity (*Patel et al., 2011*). These studies have enabled us to identify functionally distinct roles of chromatin assembly and remodeling in the formation of periodic nucleosome arrays.

## Results and discussion

### Chromatin remodeling-defective Chd1 motor proteins can assemble nucleosomes

To investigate the relation between ATP-dependent chromatin assembly and ATP-dependent chromatin remodeling, we analyzed two mutant versions of *Saccharomyces cerevisiae* Chd1 (yChd1) that exhibit substantial (~40% of wild-type) nucleosome-stimulated ATPase activity but are nearly completely deficient (<0.1% of wild-type) in chromatin remodeling activity, as assessed by the nucleosome sliding assay (*Patel et al., 2011*). These chromatin remodeling-defective yChd1 proteins contain either a deletion of residues 932–940 (Δ932–940) or a Trp$^{932}$ to Ala substitution (W932A; *Figure 1A*). The Δ932–940 and W932A mutations of yChd1 block its ability to couple ATPase activity to nucleosome

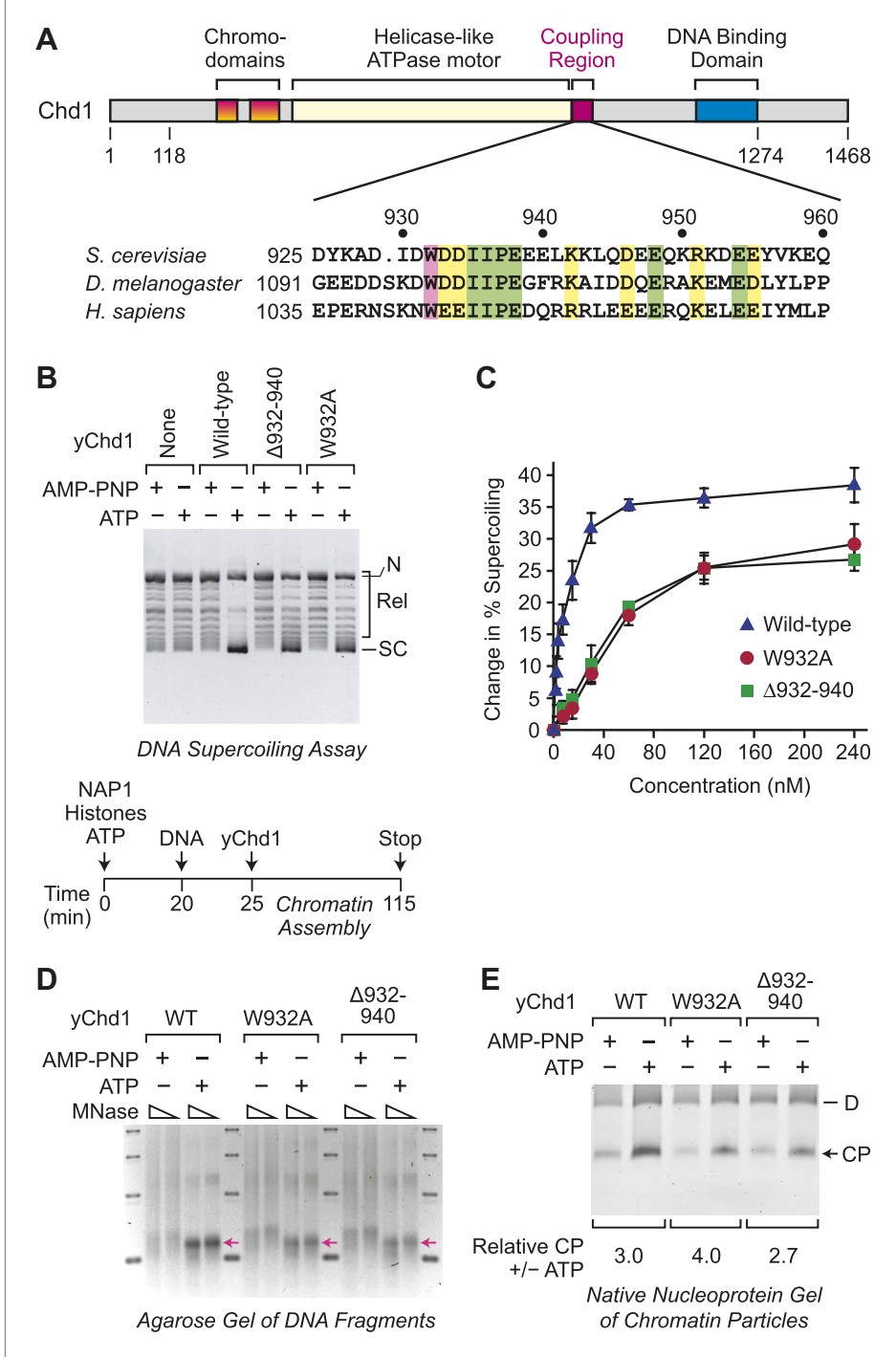

**Figure 1**. ATP-dependent nucleosome assembly is functionally distinct from chromatin remodeling. (**A**) Diagram of Chd1 and the conserved coupling region. The numbers below the schematic diagram and above the amino acid sequences indicate positions in *S. cerevisiae* Chd1 (yChd1). (**B**) Chromatin remodeling-defective mutant yChd1 proteins (yChd1Δ932–940; yChd1W932A) can assemble nucleosomes in an ATP-dependent manner. Chromatin assembly reactions were performed with wild-type or mutant yChd1 proteins (120 nM) in the presence of either adenylyl-imidodiphosphate (AMP-PNP) or ATP. The efficiency of nucleosome assembly was monitored by the DNA supercoiling assay. The positions of supercoiled (SC), relaxed (Rel), and nicked open circular (N) DNAs are indicated. (**C**) Quantitative analysis of the efficiency of nucleosome assembly by mutant vs wild-type yChd1 proteins. Chromatin assembly reactions with yChd1 proteins were analyzed by the DNA supercoiling assay. The change in % supercoiling ([Δ supercoiled DNA/total DNA] × 100%) vs concentration of yChd1 (nM) is shown. The results are presented as the
*Figure 1. Continued on next page*

*Figure 1. Continued*

mean ± standard deviation (*N* ≥ 3). (**D**) Agarose gel electrophoresis of DNA fragments derived from chromatin assembled with wild-type or mutant yChd1 proteins. Chromatin assembly reactions were carried out as in (**B**), except that the concentration of Chd1 proteins was 60 nM. The reaction products were digested extensively with micrococcal nuclease (MNase) and subsequently deproteinized. The resulting DNA fragments were resolved on a 3% agarose gel and visualized by staining with ethidium bromide. The arrows indicate the position of DNA fragments derived from core particles. (**E**) Native nucleoprotein gel analysis of nucleosomes assembled with wild-type or mutant yChd1 proteins. Chromatin assembly reactions were carried out as in (**D**). The reaction products were digested extensively with MNase; the resulting nucleoprotein complexes were subjected to electrophoresis on a nondenaturing 5% polyacrylamide gel; and the DNA was stained with Sybr Green I (Invitrogen). The positions of core particles (CP) and dinucleosomes (D) are indicated.

remodeling/sliding, but still allow for robust ATPase activity. In contrast, deletions of more C-terminal regions of yChd1 (ranging from residues 939–1010) cause the near complete loss of the ATPase activity (*Patel et al., 2011*; *Ryan et al., 2011*). Hence, for our analysis, we employed the Δ932–940 and W932A mutant yChd1 proteins.

We first examined whether the chromatin remodeling-defective yChd1 proteins are able to function in the ATP-dependent assembly of nucleosomes. To this end, we used the purified, defined ATP-dependent chromatin assembly system that consists of NAP1, core histones, Chd1, ATP, and relaxed DNA (*Lusser et al., 2005*). Chromatin assembly requires both the Chd1 motor protein as well as the NAP1 histone chaperone (*Lusser et al., 2005*). The extent of nucleosome assembly was monitored by the DNA supercoiling assay, in which the change in the linking number of DNA that occurs upon formation of nucleosomes in the presence of topoisomerase I is observed (*Germond et al., 1975*; *Simpson et al., 1985*). In these experiments, we observed that the mutant yChd1 proteins are able to assemble nucleosomes in an ATP-dependent manner (*Figure 1B*). In the presence of adenylyl imidodiphosphate (AMP-PNP), a β-γ-non-hydrolyzable analog of ATP, chromatin assembly was not observed. The extent of nucleosome assembly by the mutant proteins was about 65% of that of wild-type yChd1 (*Figure 1C*).

We further tested whether the mutant yChd1 proteins catalyze the formation of nucleosomes by micrococcal nuclease (MNase) digestion analysis. First, we extensively digested the reaction products with MNase, deproteinized the samples, and analyzed the resulting DNA fragments by agarose gel electrophoresis (*Figure 1D*). With the wild-type as well as the mutant yChd1 proteins, we observed the ATP-dependent formation of DNA fragments corresponding to the length of a core particle. However, in the absence of ATP, particularly with the mutant yChd1 proteins, there were DNA fragments that are longer than those obtained in the presence of ATP. We therefore examined whether or not the species formed in the absence of ATP contained canonical nucleosomes by nondenaturing gel electrophoresis of the MNase digestion products (*Figure 1E*). In this assay, we observed ATP-dependent stimulation of the formation of core particles by wild-type as well as mutant yChd1 proteins. Thus, the mutant yChd1 proteins catalyze the ATP-dependent formation of nucleosomes, as assessed by the generation of chromatin that yields core particles upon extensive MNase digestion.

To determine the relative rates of chromatin assembly by the wild-type and mutant yChd1 proteins, we performed kinetic analyses and found that the initial rates of nucleosome assembly by the Δ932–940 and W932A proteins were approximately 8.5% and 12% of the rate of wild-type yChd1 (*Figure 2*). Hence, the chromatin-remodeling defective yChd1 proteins exhibit substantial ATP-driven nucleosome assembly activity (~10% of the rate of wild-type yChd1) that is at least 100-fold higher than their ATP-driven chromatin remodeling/sliding activity (<0.1% of wild-type yChd1). The properties of wild-type and mutant yChd1 proteins indicate that the ATP-dependent catalysis of nucleosome assembly appears to be a functionally distinct process from the ATP-dependent remodeling of chromatin.

## Chromatin remodeling-defective Chd1 proteins do not yield periodic nucleosomes

Because ATP-dependent chromatin assembly factors such as Chd1 are able to catalyze the formation of regularly-spaced nucleosome arrays (e.g., *Ito et al., 1997*; *Varga-Weisz et al., 1997*; *Loyola et al., 2001*; *Lusser et al., 2005*; *Lewis et al., 2010*; *Emelyanov et al., 2012*), we examined the periodicity of the nucleosomes assembled by the mutant yChd1 proteins by using the partial MNase digestion

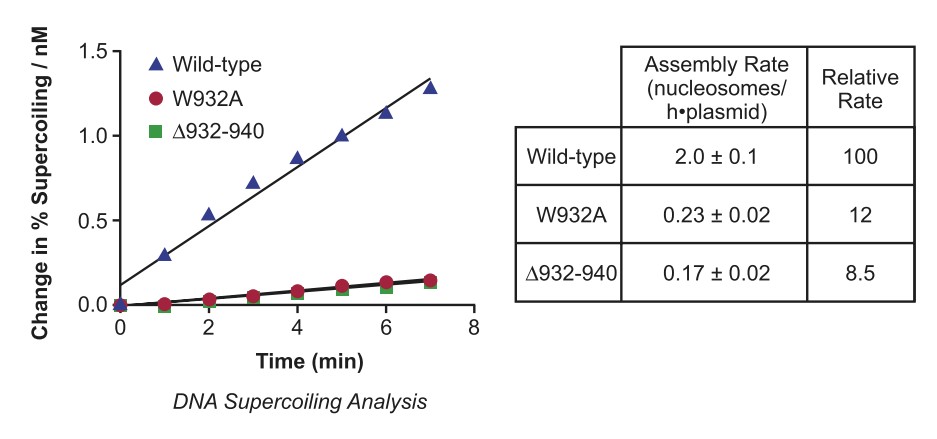

**Figure 2**. Analysis of the initial rates of nucleosome assembly by the wild-type, W932A, and Δ932–940 yChd1 proteins. The initial rates were measured as change in % supercoiling ([Δ supercoiled DNA/total DNA] × 100%)/(nM protein) vs time (min). The table summarizes the nucleosome assembly rates as mean ± standard deviation (N = 3). The relative rates are given with respect to that of the wild-type protein.

assay. These experiments revealed that the chromatin remodeling-defective yChd1 proteins are unable to generate periodic arrays of nucleosomes (*Figure 3*). However, because the mutant yChd1 proteins are not fully active for nucleosome assembly (*Figures 1 and 2*), it was difficult to attribute the absence of periodic nucleosome arrays in the MNase assay (*Figure 3*) to decreased efficiency of assembly or to a defect in the formation of periodic nucleosome arrays.

To determine the ability of the mutant yChd1 proteins to yield periodic nucleosome arrays, we employed a nucleosome spacing assay, which is depicted in *Figure 4A*. In this assay, the extent of ATP-dependent conversion of randomly-distributed nucleosomes (formed by the salt-dialysis method) to evenly-spaced nucleosome arrays was monitored by the partial MNase digestion assay. As seen in *Figure 4B*, the mutant yChd1 proteins are defective in the ATP-dependent formation of periodic nucleosome arrays. To quantitate the ATP-dependent catalysis of nucleosome spacing, we devised a

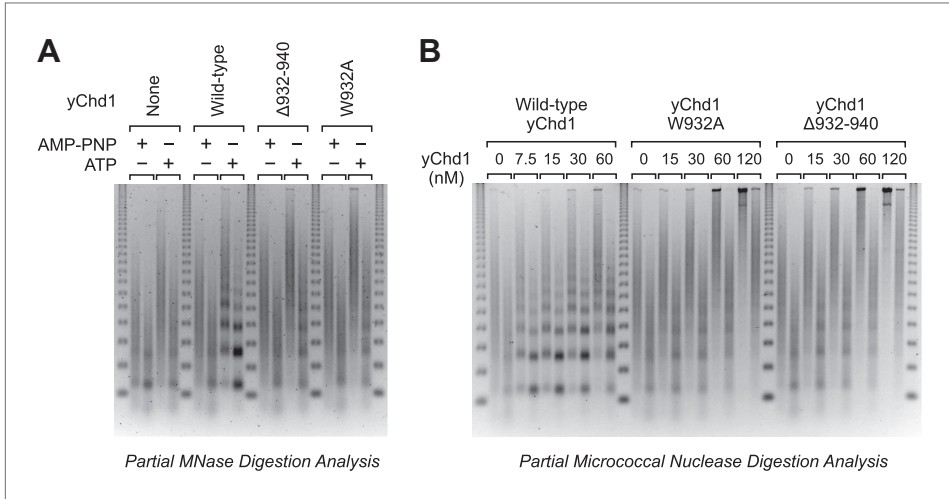

**Figure 3**. Chromatin remodeling-defective yChd1 proteins are unable to generate arrays of evenly-spaced nucleosomes during chromatin assembly. (**A**) Chromatin assembly reactions were performed with wild-type or mutant yChd1 proteins (30 nM) in the presence of AMP-PNP or ATP. The reaction products were subjected to partial MNase digestion analysis. (**B**) Chromatin assembly reactions were performed with the indicated concentrations of yChd1 proteins and then subjected to partial MNase digestion analysis.

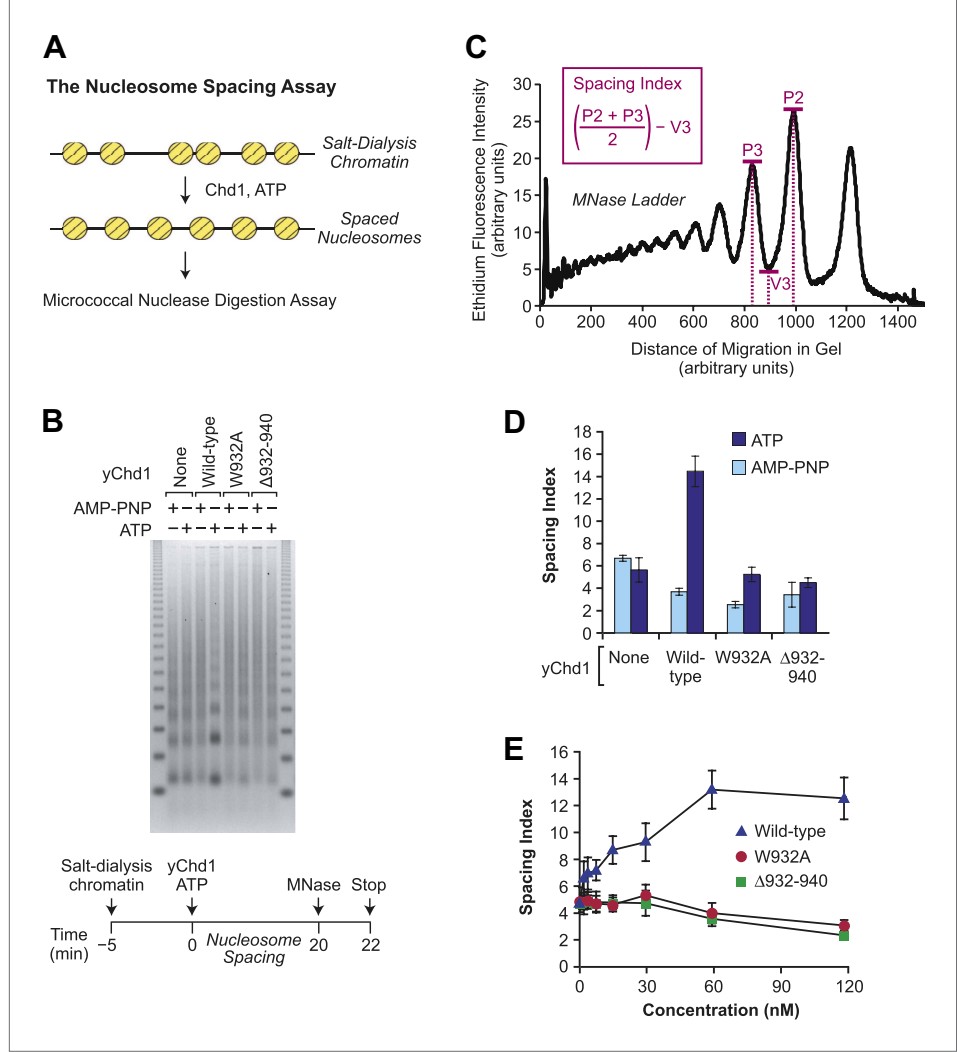

**Figure 4**. The chromatin remodeling-defective yChd1 proteins are not able to catalyze the formation of regularly-spaced nucleosomes. (**A**) Schematic representation of the nucleosome spacing reaction. Randomly-distributed nucleosomes were generated by salt dialysis reconstitution of chromatin. Wild-type or mutant Chd1 protein was added along with ATP, and the reaction products were characterized by partial MNase digestion analysis. (**B**) Mutant yChd1 proteins do not reposition nucleosomes into periodic arrays. Spacing reactions were performed by incubating salt-dialysis chromatin with wild-type or mutant yChd1 proteins (30 nM) in the presence of either AMP-PNP or ATP. The reaction products were characterized by partial MNase digestion analysis. (**C**) Determination of the spacing index. Agarose gels from nucleosome spacing reactions were stained by ethidium bromide and then subjected to imaging and analysis on ImageQuantTL (GE) to obtain densiometry scans. The spacing index is the average height of the di- and tri-nucleosome peaks ([P2 + P3]/2) minus the height of the valley (V3) between the peaks, as indicated by the formula shown in the figure. (**D**) Quantitative analysis of nucleosome spacing by wild-type and mutant yChd1 proteins. The spacing indices were determined for the products of reactions such as those shown in (**B**). The results are presented as the mean ± standard deviation ($N = 6$). (**E**) The chromatin remodeling-defective yChd1 proteins exhibit little to no activity in the nucleosome spacing assay at different concentrations. Nucleosome spacing reactions were performed with yChd1 proteins and subjected to partial MNase digestion analysis. The graph depicts the average spacing index ± standard deviation ($N = 4$) vs concentration (nM) for each of the indicated proteins.

---

nucleosome spacing index (*Figure 4C*) that measures the formation of distinct di- and tri-nucleosome MNase digestion bands. The spacing index increases with the periodicity of the chromatin, and can be used for the comparison of series of samples that are under identical electrophoretic conditions. For example, quantitation of the nucleosome spacing data in *Figure 4B* reveals that wild-type yChd1 has

strong ATP-dependent spacing activity, whereas the mutant yChd1 proteins exhibit little or no spacing activity (*Figure 4D*). The essentially complete absence of spacing activity in the mutant yChd1 proteins was further observed in experiments that were carried out with concentrations of yChd1 proteins ranging from 2–120 nM (*Figure 4E*). (Examination of the spacing index also reveals that the periodicity of the partial MNase digestion array slightly decreases upon addition of Chd1 proteins in the presence of AMP-PNP. This effect may be due to the blockage of MNase digestion by static nonproductive binding of the Chd1 to DNA.) Hence, the mutant yChd1 proteins are able to catalyze the ATP-dependent assembly of nucleosomes, but are not able to mediate the ATP-dependent formation of regularly-spaced nucleosomes.

## The distinction between chromatin assembly and remodeling by Chd1 is conserved

To determine whether the functionally distinct chromatin assembly and remodeling functions of Chd1 are conserved from yeast to metazoans, we generated and purified mutant versions of *Drosophila melanogaster* Chd1 (dChd1) that correspond to the mutant yChd1 proteins (*Figures 1A and 5A*). Specifically, dChd1 Δ1099–1107 is analogous to yChd1 Δ932–940, and dChd1 W1099A is analogous to yChd1 W932A. Similar to that seen with the yChd1 proteins, the nucleosome-stimulated ATPase activities of the mutant dChd1 proteins are 37–38% of the wild-type activity, whereas the ATP-dependent chromatin remodeling activity of each of the mutant dChd1 proteins is about 2% of that of the wild-type protein (*Table 1*). The chromatin remodeling-defective dChd1 proteins exhibit substantial (~65% of wild-type) ATP-dependent nucleosome assembly activity, as measured by the DNA supercoiling assay (*Figure 5B*), but are not able to convert naked DNA into periodic nucleosome arrays in chromatin assembly assays (*Figure 5C*) or to catalyze the ATP-dependent spacing of nucleosome (*Figure 5D*). The magnitudes of the differences between the assembly and remodeling activities of the wild-type vs mutant dChd1 proteins is not as large as those seen with yChd1. Nevertheless, it is evident that the functional distinction between nucleosome assembly and remodeling/sliding/spacing in Chd1 is conserved from yeast to *Drosophila*.

## ATP-dependent chromatin remodeling is not sufficient for nucleosome assembly

The analysis of the mutant Chd1 proteins indicated that ATP-dependent chromatin assembly does not require chromatin remodeling activity. It remained formally possible, however, that chromatin remodeling is sufficient for chromatin assembly. To test this notion, we carried out parallel chromatin remodeling and assembly reactions with purified yChd1 (as a positive control/reference) and purified human Brg1 (hBrg1; also known as SMARCA4), which is a SNF2-like family ATPase found in human SWI/SNF chromatin remodeling complexes (reviewed in *Clapier and Cairns, 2009*; *Flaus and Owen-Hughes, 2011*; *Hargreaves and Crabtree, 2011*; *Ryan and Owen-Hughes, 2011*). The purified hBrg1 protein is active for chromatin remodeling (e.g., *Phelan et al., 1999*). By using the restriction enzyme accessibility assay for chromatin remodeling (e.g., as in *Varga-Weisz et al., 1997*; *Boyer et al., 2000*; *Shen et al., 2000*; *Alexiadis and Kadonaga, 2002*), we found that remodeling activity of yChd1 was comparable to that of hBrg1 over a range of concentrations (*Figure 6A*). We then performed chromatin assembly reactions with the same concentrations of factors, and found that hBrg1 did not assemble chromatin under conditions in which efficient chromatin assembly was observed with the yChd1 control (*Figure 6B*). Moreover, the hBrg1 caused a decrease in the supercoiling of DNA, which may be related to the nucleosome disruption activity of the SWI/SNF complex (e.g., *Kwon et al., 1994*). These results thus show that ATP-dependent chromatin remodeling does not necessarily result in the formation of nucleosomes from histones, histone chaperone (NAP1), DNA, and ATP.

## Two ATP-dependent processes yield periodic nucleosome arrays during assembly

In the context of chromatin assembly, our findings suggest that two ATP-dependent processes are involved in the formation of periodic arrays of nucleosomes (*Figure 7*). First, an ATP-driven chromatin assembly activity generates randomly-distributed nucleosomes from histones, histone chaperone(s), and DNA, probably via a prenucleosome intermediate (*Torigoe et al., 2011*). Then, the randomly-distributed nucleosomes are converted into periodic nucleosome arrays via an ATP-driven nucleosome spacing (remodeling) activity. Although we depict the two processes separately, they may occur concurrently with wild-type Chd1. The basis for the formation of periodic nucleosomes as the final

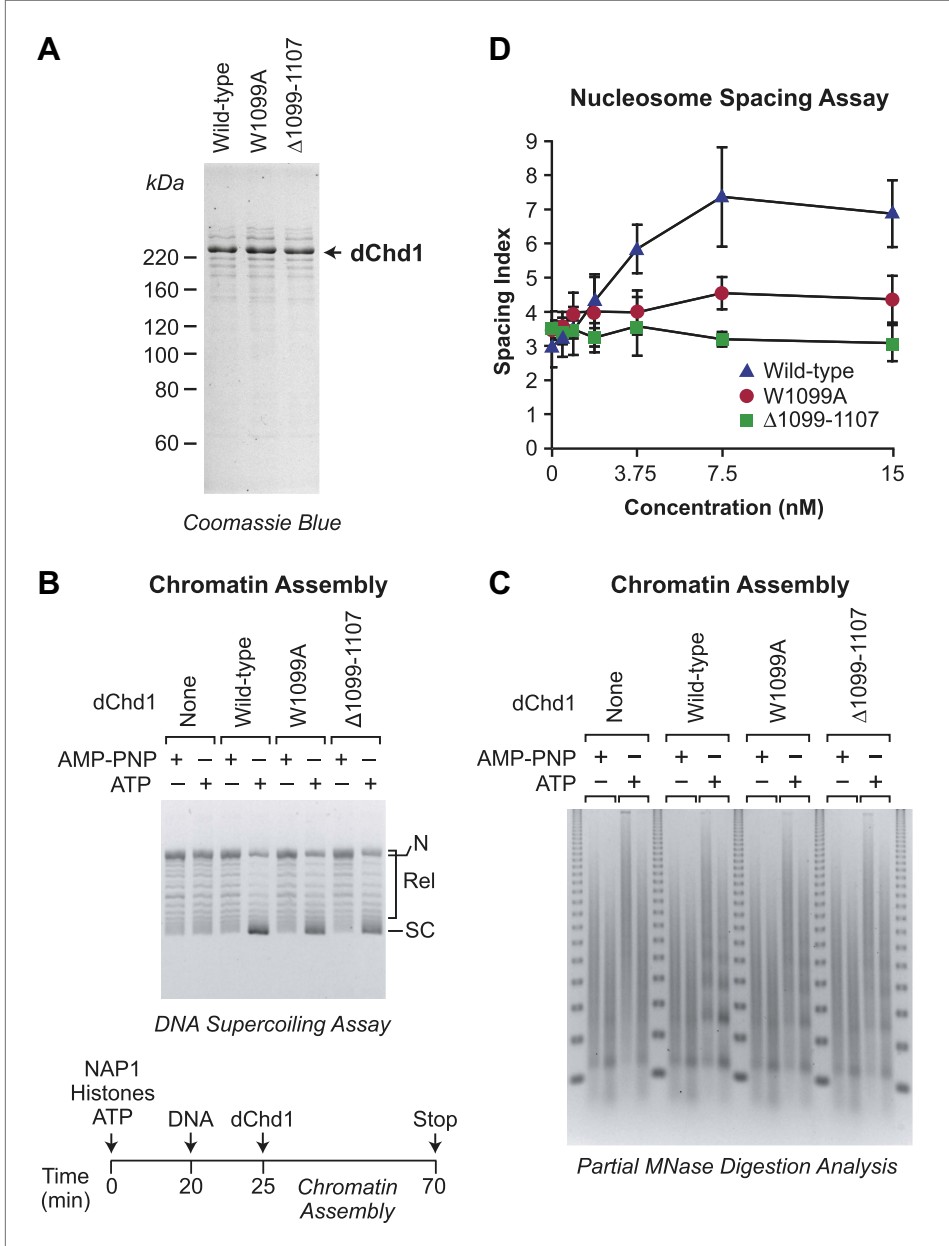

**Figure 5**. ATP-dependent nucleosome assembly is observed with chromatin remodeling-defective *Drosophila* Chd1 (dChd1) proteins. (**A**) Purification of recombinant wild-type and mutant dChd1. The purified proteins were analyzed by polyacrylamide-SDS gel electrophoresis and visualized by staining with Coomassie Blue. (**B**) Chromatin remodeling-defective dChd1 proteins can assemble nucleosomes in an ATP-dependent manner. Chromatin assembly reactions were performed with wild-type or mutant dChd1 proteins (60 nM) in the presence of either AMP-PNP or ATP. The reaction products were analyzed by the DNA supercoiling assay. The positions of supercoiled (SC), relaxed (Rel), and nicked open circular (N) DNAs are indicated. (**C**) Chromatin assembly with remodeling-defective dChd1 proteins does not yield regularly-spaced nucleosomes. Chromatin assembly reactions were performed with wild-type or mutant dChd1 proteins (60 nM) in the presence of either AMP-PNP or ATP. Reaction products were subjected to partial MNase digestion analysis. (**D**) The chromatin remodeling-defective dChd1 proteins exhibit reduced nucleosome spacing activity. Spacing assays were performed with dChd1 proteins and subjected to partial MNase digestion analysis. This graph depicts the mean spacing index ± standard deviation (*N* = 3) vs concentration (nM).

**Table 1.** Nucleosome sliding and ATP hydrolysis activities of wild-type and mutant dChd1 proteins

| dChd1 protein | Nucleosome sliding activity | | ATP hydrolysis activity (stimulated by DNA) | | ATP hydrolysis activity (stimulated by nucleosomes) | |
|---|---|---|---|---|---|---|
| | $k_{cat}$ (min⁻¹) | Relative rate (%) | $k_{cat}$ (min⁻¹) | Relative rate (%) | $k_{cat}$ (min⁻¹) | Relative rate (%) |
| Wild-type | 2.2±0.2 | 100 | 61±2 | 100 | 115±3 | 100 |
| W1099A | 0.045±0.009 | 2.0±0.4 | 34±2 | 56±3 | 44±3 | 38±3 |
| Δ1099–1107 | 0.039±0.011 | 1.8±0.5 | 30±2 | 49±3 | 42±4 | 37±3 |

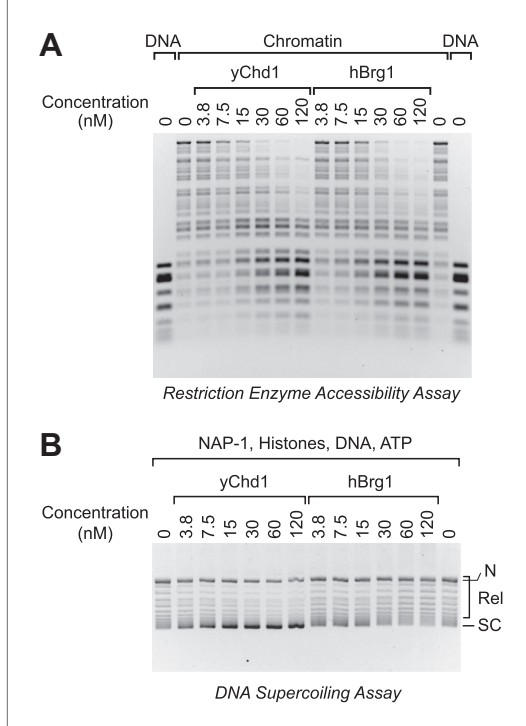

Figure 6. The Brg1 chromatin remodeling factor does not catalyze chromatin assembly. (**A**) Purified human Brg1 (hBrg1) has a specific activity for chromatin remodeling that is similar to that of wild-type yChd1. Restriction accessibility assays were performed with salt dialysis-reconstituted chromatin with Hae III restriction enzyme and the indicated concentrations of either yChd1 or hBrg1. Naked DNA was used as a reference. After digestion with Hae III, the nucleic acids were deproteinized, subjected to agarose gel electrophoresis, and then visualized by staining with ethidium bromide. (**B**) Brg1 does not assemble chromatin. Chromatin assembly reactions were performed with the indicated concentrations of yChd1 or hBrg1, and the extent of chromatin assembly was monitored by the DNA supercoiling assay.

product of chromatin assembly is not known, but it is possible that attractive forces between nucleosomes are maximized when the nucleosomes are arranged and/or compacted in a periodic array. In addition, the internucleosomal spacing may be influenced by the interaction of the factors with DNA.

In this process, it might be expected that the individual rates of nucleosome assembly and nucleosome spacing would be at least as fast as the overall rate of assembly of periodic nucleosome arrays. We therefore compared the relative rates of the following: (i) nucleosome assembly (from naked DNA to randomly-distributed nucleosomes; first step in *Figure 7*); (ii) nucleosome spacing (from randomly-distributed nucleosomes to periodic nucleosomes; second step in *Figure 7*); and (iii) the overall assembly of naked DNA into periodic nucleosome arrays (both steps in *Figure 7*). We performed each of these processes under identical conditions (*Figure 8*), and found that the individual rates of nucleosome assembly and nucleosome spacing are indeed faster than the overall rate of assembly of spaced nucleosomes. Hence, the chromatin assembly process depicted in *Figure 7* is compatible with the kinetic data.

## Summary and perspectives

In this study, we observed a conserved functional distinction between ATP-dependent nucleosome assembly and ATP-dependent chromatin remodeling by Chd1. Specifically, the ATP-dependent assembly of nucleosomes by Chd1 occurs in the near complete absence of ATP-dependent chromatin remodeling. Hence, these findings revealed a level of specificity in the role of ATP-dependent motor proteins in chromatin assembly that had not been previously anticipated. In addition, the ATP-dependent chromatin remodeling protein, human Brg1, does not assemble nucleosomes. We further examined the ability of the chromatin remodeling-defective Chd1 proteins to function in the assembly of periodic nucleosome arrays. The results suggest a process (*Figure 7*) in which randomly-distributed canonical nucleosomes are generated by the nucleosome assembly function of Chd1 and are then

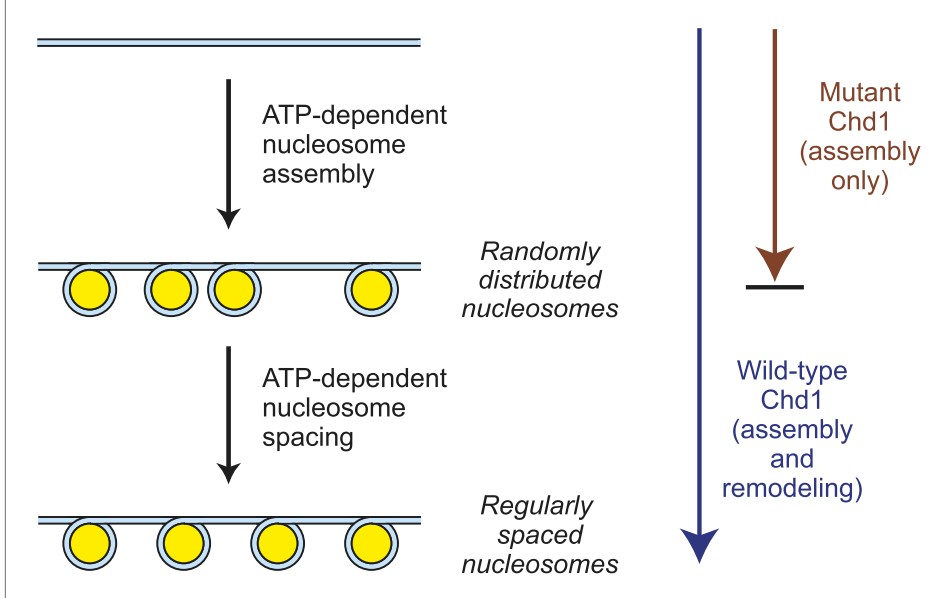

**Figure 7**. A model for the functions of ATP-driven nucleosome assembly and remodeling activities in the assembly of chromatin. In this model, nucleosomes can be assembled by an ATP-dependent motor protein in the essentially complete absence of chromatin remodeling activity, as seen with the mutant Chd1 proteins. The resulting nucleosomes are, however, randomly distributed throughout the DNA template. These randomly-distributed nucleosomes can be converted into periodic nucleosome arrays by an ATP-dependent nucleosome remodeling (spacing) activity that can be distinguished from the chromatin assembly activity. These processes may occur concurrently with the wild-type Chd1.

converted into regularly-spaced nucleosome arrays by the chromatin remodeling/spacing/sliding activity of Chd1. This model for chromatin assembly should provide a useful framework for the analysis of the regeneration of nucleosomes during the many processes in the eukaryotic nucleus that involve the disassembly and reassembly of chromatin.

## Materials and methods

### Protein purification

*D. melanogaster* NAP1 and topoisomerase I (ND432 N-terminally truncated form containing the catalytic domain) were purified as described (*Fyodorov and Kadonaga, 2003*). Native *D. melanogaster* core histones were purified from embryos by the method of *Fyodorov and Levenstein (2002)*. Wild-type and mutant (internal deletion Δ932–940 and point mutant W932A) forms of truncated *S. cerevisiae* Chd1 (amino acids 118–1274), which possesses the core chromodomains, ATPase motor, and DNA-binding domain, were synthesized in bacteria and purified, as described (*Patel et al., 2011*). Human Brg1 was purified as described previously (*Phelan et al., 1999*).

The coding sequence for full-length *D. melanogaster* Chd1 was subcloned into pDEST17 vectors (Invitrogen, Carlsbad, CA). The internal deletion (Δ1099–1107) and single amino acid substitution (W1099A) constructs were generated by PCR. The *D. melanogaster* Chd1 proteins were synthesized in *Escherichia coli* BL21-star (DE3) cells (Invitrogen) containing two additional plasmids—a trigger factor chaperone overexpression plasmid (Li Ma and Guy Montelione, Rutgers University) and the RIL plasmid for rare tRNAs (Stratagene, Santa Clara, CA). Two liters of bacterial culture were grown at 37°C to $A_{600}$ ~0.5–0.6 and induced with 0.5 mM IPTG. The cells were incubated at 17°C for 18 hr, harvested, and sonicated in Buffer A (50 mM Tris-HCl, pH 7.5, 2 mM $MgCl_2$, 500 mM NaCl, 10% [vol/vol] glycerol, 10 mM β-mercaptoethanol, 1 mM benzamidine, and 0.2 mM PMSF) containing 10 mM imidazole. The lysate was cleared by centrifugation (20 min; 45,000×*g*; 4°C) and mixed with 3 ml Ni-NTA agarose (Qiagen, Germantown, MD). After incubation on a rotating wheel for 3 hr at 4°C, the resin was loaded into a disposable 20-ml polypropylene column and washed with 30 column volumes of Buffer A containing 10 mM imidazole. His-tagged Chd1 was then eluted with 10 ml of Buffer A containing 300 mM imidazole. Following concentration with an

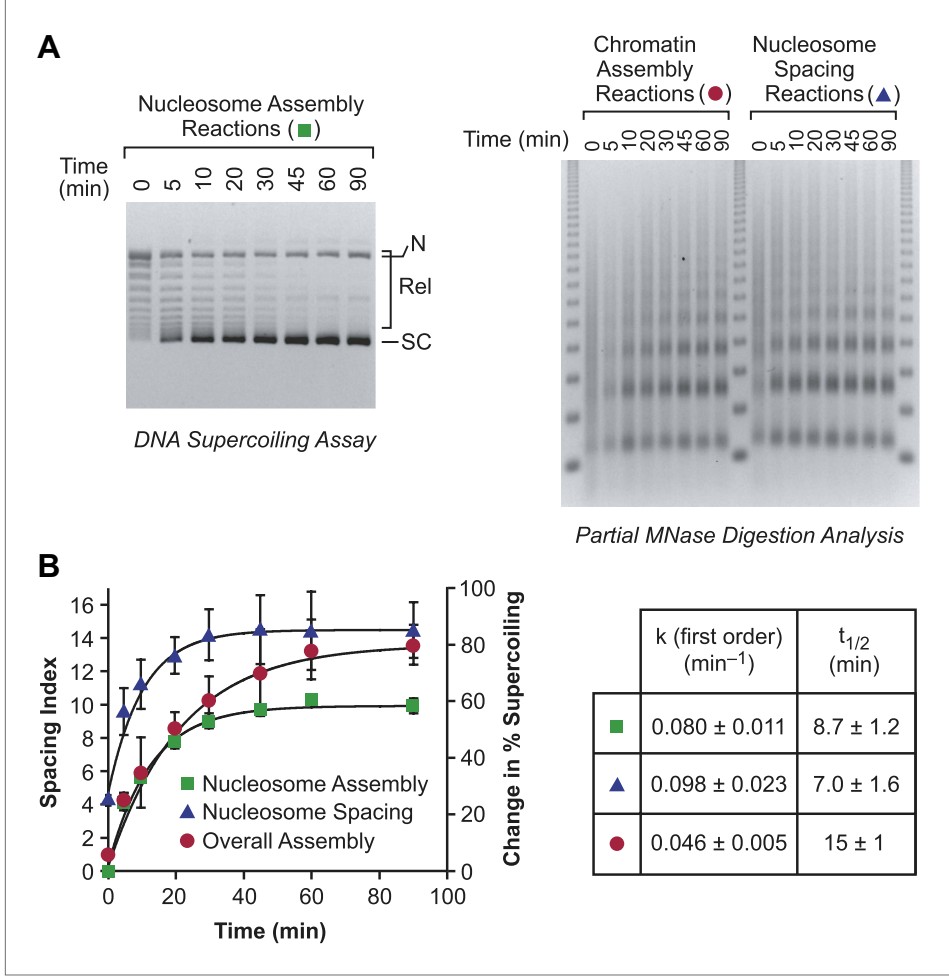

**Figure 8**. The individual rates of nucleosome formation and spacing are faster than the overall rate of assembly of periodic arrays of nucleosomes. (**A**) Determination of the rates of nucleosome formation, nucleosome spacing, and assembly of regularly-spaced nucleosomes. Nucleosome formation in chromatin assembly reactions was monitored by using the DNA supercoiling assay. The positions of supercoiled (SC), relaxed (Rel), and nicked open circular (N) DNAs are indicated. Nucleosome spacing reactions were analyzed by partial MNase digestion analysis. The overall assembly of periodic arrays of nucleosomes was determined by performing chromatin assembly reactions and analyzing the reaction products by partial MNase digestion analysis. All reactions were performed with wild-type yChd1 at 30 nM. (**B**) Quantitation of the nucleosome formation and spacing assays, such as those shown in (**A**). The figure displays the change in % supercoiling ([$\Delta$ supercoiled DNA/total DNA] $\times$ 100%) and spacing indices vs reaction time (min). The reactions followed first-order kinetics, and the first-order rate constants (min$^{-1}$) and half-times for reaction (t$_{1/2}$, min) are given in the table. The data points are presented as mean ± standard deviation (N = 3), and are depicted with the plots of the first order curves (r$^2$ > 0.95 for all graphs).

Amicon Ultra-15 (30 kDa nominal molecular weight limit) Centrifugal Filter Unit (Millipore, Billerica, MA), Chd1 was applied to a Superdex 200 prep grade (GE Healthcare, Piscataway, NJ) size exclusion column ([column volume, 120 ml]; column dimensions [diameter × length], 1.6 cm × 60 cm; flow rate, 1.0 ml/min) and eluted with 1.5 column volumes of Buffer B (50 mM Tris-HCl, pH 7.5, 2 mM MgCl₂, 10% [vol/vol] glycerol, 10 mM β-mercaptoethanol, 1 mM benzamidine, and 0.2 mM PMSF) containing 300 mM NaCl. The peak fractions were analyzed by polyacrylamide-SDS gel electrophoresis, pooled, and dialyzed against Buffer B containing 100 mM NaCl. The resulting sample was applied to a Source 15S (GE Healthcare) cation exchange column ([column volume, 1.0 ml]; column dimensions [diameter × length], 0.5 cm × 5 cm; flow rate, 1.0 ml/min). The column was washed with 10 column volumes of 100 mM NaCl in Buffer B, and the protein was eluted with a linear gradient (12 column volumes) from 100 mM to 1 M NaCl in Buffer B. Peak fractions were pooled and dialyzed against Buffer B containing 50 mM NaCl.

## Chromatin assembly

Chromatin assembly reactions were performed as described previously (*Fyodorov and Kadonaga, 2003*; *Lusser et al., 2005*). All reactions contained core histones (0.353 µg), NAP1 (1.4 µg), relaxed circular DNA plasmid (0.294 µg), ATP (3 mM), topoisomerase I (1 nM), and an ATP regeneration system (3 mM phosphoenolpyruvate, 20 U/µl pyruvate kinase) in a final volume of 70 µl. The buffer composition of the final reaction mixture was as follows: 15 mM Hepes (K+), pH 7.6, 3 mM Tris, 100 mM KCl, 5 mM NaCl, 5.5 mM $MgCl_2$, 0.1 mM EDTA, 6.6% (vol/vol) glycerol, 1% (wt/vol) polyvinyl alcohol (average MW 10,000), 1% (wt/vol) polyethylene glycol 8000, and 20 µg/ml bovine serum albumin. The reaction products were analyzed by DNA supercoiling and partial MNase digestion assays (*Fyodorov and Kadonaga, 2003*) as well as by extensive MNase digestion of the reaction products followed by agarose gel electrophoresis of DNA fragments (e.g., *Torigoe et al., 2011*) or native nucleoprotein gel electrophoresis of chromatin particles (*Varshavsky et al., 1976*). The percent supercoiling ([amount of supercoiled DNA/amount of total DNA species] × 100%) in the DNA supercoiling assays was quantified with ImageQuantTL (GE Healthcare). Because it is not possible to ascertain the fraction of nicked DNA that is packaged into chromatin, we included the nicked DNA in the 'amount of total DNA species' but not in the 'amount of supercoiled DNA'. Therefore, 'percent supercoiling' reflects the amount of closed circular plasmid DNA that is packaged into chromatin and does not include the nicked DNA that is packaged into chromatin.

## Nucleosome spacing assays

Nucleosomes were reconstituted onto plasmid DNA by the salt-dialysis method and purified by sucrose gradient sedimentation. The resulting chromatin (0.147 µg of DNA) was incubated with topoisomerase I (1 nM), ATP (3 mM), an ATP regeneration system (3 mM phosphoenolpyruvate, 20 U/µl pyruvate kinase), and Chd1 proteins in a final volume of 35 µl. The reaction medium was identical to that used for chromatin assembly. The reaction products were analyzed by partial MNase digestion assays (*Fyodorov and Kadonaga, 2003*).

## Calculation of the spacing index

Chromatin assembly and nucleosome spacing reactions were subjected to partial MNase digestion analysis. Gels were imaged and analyzed with ImageQuantTL (GE Healthcare) to obtain densitometry scans of the ethidium-stained bands. Following subtraction of background staining, three heights in the signal were determined: the maximum of the peak corresponding to dinucleosomes (P2), the maximum of the peak corresponding to trinucleosomes (P3), and the minimum of the valley between these two peaks (V2). The spacing index was calculated by using the following equation: (0.5)(P2 + P3) − V2.

## Nucleosome sliding assays

Nucleosomes were reconstituted by the gradient salt dialysis method by using *S. cerevisiae* core histones and a FAM-labeled 208 bp fragment with a 601 nucleosome positioning sequence at one end (*Patel et al., 2011*). Nucleosomes were purified over a mini prep-cell (Bio-Rad, Hercules, CA). Sliding reactions, which monitor the Chd1-catalyzed movement of nucleosomes on a 208 bp DNA fragment, were performed as previously described (*Patel et al., 2011*), with 0-N-63 nucleosomes (100 nM) incubated with dChd1 proteins (100 nM) in buffer containing 20 mM Hepes-K+, pH 7.6, 50 mM KCl, 1 mM DTT, 0.1 mM EDTA, 5% sucrose, 0.1 mg/ml BSA, 2.5 mM ATP, and 5 mM $MgCl_2$. Reactions were carried out at 23°C and quenched at the indicated times with a stop solution containing 25 mM EDTA and 2 mg/ml DNA. Changes in nucleosome positions over time were resolved by native polyacrylamide gels, and quantified with ImageJ. Data are averages of three or more separate experiments, and sliding rates were calculated from single exponential fits to data.

## ATP hydrolysis assay

ATPase rates were determined using an NADH-coupled assay as previously described (*Patel et al., 2011*). Briefly, dChd1 proteins (50 nM) were incubated in the absence or presence of DNA or nucleosome substrates up to 500 nM concentration. Substrates were the same 208 bp DNA fragment either alone or reconstituted into nucleosomes with yeast histones. Data were fit to the Michalis-Menton equation in Kaleidagraph, $k_{obs} = (k_{cat})[S]/(Km + [S])$, where [S] is the initial concentration of substrate. The DNA- and nucleosome-stimulated rate constants were subtracted from the basal rate constants in the absence of substrates, which were on the order of 10–30 $min^{-1}$.

## Restriction enzyme accessibility assay

Restriction enzyme accessibility assays were performed as described previously (*Alexiadis and Kadonaga, 2002*; *Rattner et al., 2009*).

## Acknowledgements

We thank Alexandra Lusser, Timur Yusufzai, George Kassavetis, Yuan Wang, James Gucwa, Sascha Duttke, and Barbara Rattner for critical reading of the manuscript. We thank Li Ma and Guy Montelione for chaperone expression plasmids. SET was partially supported by the Chancellor's Interdisciplinary Collaboratories Program at University of California, San Diego.

## Additional information

### Competing interests

JTK: Reviewing editor, *eLife*. The other authors declare that no competing interests exist.

### Funding

| Funder | Grant reference number | Author |
| --- | --- | --- |
| National Institute of General Medical Sciences, National Institutes of Health | GM058272 | James T Kadonaga |
| National Institute of General Medical Sciences, National Institutes of Health | GM084192 | Gregory D Bowman |

The funder had no role in study design, data collection and interpretation, or the decision to submit the work for publication.

### Author contributions

SET, Conception and design, Acquisition of data, Analysis and interpretation of data, Drafting or revising the article; AP, Conception and design, Acquisition of data, Analysis and interpretation of data; MTK, Acquisition of data, Analysis and interpretation of data, Drafting or revising the article; GDB, JTK, Conception and design, Analysis and interpretation of data, Drafting or revising the article

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
