## [Decision Letter]

Thank you for sending your work entitled “ATP-dependent Chromatin Assembly Is Functionally Distinct from Chromatin Remodeling” for consideration at *eLife*. Your article has been favorably evaluated by a Senior editor and 3 reviewers, one of whom is a member of our Board of Reviewing Editors.

The Reviewing editor and the other two reviewers discussed their comments before we reached a decision, and the Reviewing editor has assembled the following comments to help you prepare a revised submission.

We believe the work will advance the field and thus we invite you to resubmit a revised manuscript addressing the following issues:

Though the Chd1 Δ932-940 and W932A mutants are capable of hydrolyzing ATP, they essentially function as ‘histone chaperones’ of a sort since they are defective for chromatin remodeling. However, the authors suggest that ATP accentuates Chd1 chromatin assembly in these mutants, yet these assays include the NAP1 histone chaperone. In spite of having BRG1 as a control, the idea that the chromatin assembly function of CHD1 by itself requires ATP may need further verification. Though careful titration of histones would be needed, would the absence of NAP1 in the actual Chd1 chromatin assembly assays still result in higher chromatin assembly levels in the presence of ATP? Would the levels still be different between the wild-type and mutant proteins? What if these mutants were combined with ATPase-defective mutations? Would any chromatin assembly be observed if NAP1 is not present? The usage of ATP could really differentiate the chromatin assembly mechanisms per se of ATP-dependent chromatin remodelers and histone chaperones, and the question is therefore important.

Additionally, nucleosome assembly is monitored solely by the change in negative supercoiling in 1D gels. Thus, it is not clear how many nucleosomes are being assembled in these reactions. Typically 2D gels are the gold standard for assembly reactions, and if a 2D gel system was used, the authors could actually quantify the number of nucleosomes/nM Chd1/min; numbers that are more informative and accurate.

With the exception of the 1D supercoiling analysis, there is little evidence that the altered Chd1 is actually assembling nucleosomes. In the MNase digestion assays, there is virtually no difference in the MNase ladders -/+ Chd1. It looks like chaperone by itself. In fact, in some cases the chaperone only lanes produce fairly good tri-nucleosome arrays. Even with Chd1, there is quite a bit of subnucleosomal product generated at fairly low MNase concentrations. The authors should present a longer MNase digestion series until the limit digest is observed.

[Editors’ note: the authors requested clarification from the editors, which is shown below.]

1) First, the manuscript (and title) seem to imply at times that CHD1 alone has chromatin assembly functions, hence comment 1. The authors should simply emphasize throughout the text that the CHD1 chromatin assembly function requires a histone chaperone and cannot be performed in vitro by CHD1 alone. With that information in mind, it will be pointless for them to repeat CHD1 chromatin assembly with mutant CHD1.

2) Regarding the information garnered from 1D vs. 2D gels. Though 2D gels may provide more information, you are right in saying it is unlikely to change the conclusion of this manuscript. It may however help with the following comment.

3) Unfortunately, the reviewer is correct in stating that the main evidence suggesting chromatin assembly in mutant CHD1 solely relies on the DNA supercoiling analyses. Though technically challenging, a second technique unambiguously concluding the same therefore remains high desirable.

---

## [Author Response]

*Though the Chd1 Δ932-940 and W932A mutants are capable of hydrolyzing ATP, they essentially function as ‘histone chaperones’ of a sort since they are defective for chromatin remodeling. However, the authors suggest that ATP accentuates Chd1 chromatin assembly in these mutants, yet these assays include the NAP1 histone chaperone. In spite of having BRG1 as a control, the idea that the chromatin assembly function of CHD1 by itself requires ATP may need further verification. Though careful titration of histones would be needed, would the absence of NAP1 in the actual Chd1 chromatin assembly assays still result in higher chromatin assembly levels in the presence of ATP? Would the levels still be different between the wild-type and mutant proteins? What if these mutants were combined with ATPase-defective mutations? Would any chromatin assembly be observed if NAP1 is not present*?

Our interpretation of these comments is that we need to test the NAP-independent chromatin assembly function of Chd1 in the presence or absence of both ATP and NAP1. With regard to this point, I should mention that we did publish the chromatin assembly activity of wild-type *Drosophila* Chd1 by itself (no NAP1) in [24] [attached; see Figures 2 and 4]. We found that wild-type Chd1 does not exhibit chromatin assembly activity without NAP1, even in the presence of ATP. We can repeat the same experiments with the mutant Chd1 proteins in the absence vs. presence of NAP1 and ATP, if desired. Given, however, that the wild-type protein lacks any detectable chromatin assembly activity in the absence of NAP1, it does seem unlikely that the mutant proteins will somehow have gained this activity. Nevertheless, if the reviewers wish for us to perform this experiment, we will.

*The usage of ATP could really differentiate the chromatin assembly mechanisms of ATP-dependent chromatin remodelers and histone chaperones, and the question is therefore important*.

Since we have previously shown that Chd1 alone (no NAP1) does not assemble chromatin (24), doesn't that already demonstrate a difference between the ATP-dependent motor proteins and histone chaperones? Also, in the present manuscript, we demonstrated the ATP dependence of Chd1 function by the use of ATP vs. AMP-PMP.

The main point of this paper is that ATP-dependent chromatin assembly is functionally distinct from ATP-dependent chromatin remodeling, as in the title. In addition, wild-type Chd1 by itself (no NAP1) does not assemble chromatin in the presence of ATP. Therefore, it is difficult to see how characterization of the ATP-dependence of the wild-type and mutant Chd1 proteins by themselves (no NAP1) would affect the conclusion of this work. Perhaps some clarification would be useful here.

*Additionally, nucleosome assembly is monitored solely by the change in negative supercoiling in 1D gels. Thus, it is not clear how many nucleosomes are being assembled in these reactions. Typically 2D gels are the gold standard for assembly reactions, and if a 2D gel system was used, the authors could actually quantify the number of nucleosomes/nM Chd1/min; numbers that are more informative and accurate*.

Over the past 24 years, we have used 2D gels on many occasions in other papers to analyze the properties of chromatin with up to four samples at a time (but typically just one or two samples). Given the limit of about four samples per gel, we would need to run samples in one series of experiments on multiple different gels. It would be difficult and essentially impossible to obtain reliable data for the comparison of different samples on different gels. Even within a single 2D gel there can be some variations due to the relatively long distances the DNA samples travel in two different dimensions. In other words, it just doesn't seem realistic or feasible to run all of the supercoiling samples on 2D gels. In our experience, we can generate the most consistent and reliable data for the comparison of a series of multiple samples with 1D gels. If our experiments had involved the quantitative analysis of two or three samples, we would have used 2D gels as we have in the past. In other words, while 2D gels can provide information on the relative amounts of differently supercoiled species of a plasmid, we do not recommend their use in the comparative analysis of a series of samples.

If we did the 2D gels, we might get different numbers, but I would not be confident that they would be more accurate or more informative. In addition, I do not really see how obtaining the data in units of number of nucleosomes/nM Chd1/min will change the conclusions of this paper.

*With the exception of the 1D supercoiling analysis, there is little evidence that the altered Chd1 is actually assembling nucleosomes. In the MNase digestion assays, there is virtually no difference in the ladders -/+ Chd1. It looks like chaperone by itself. In fact, in some cases the chaperone only lanes produce fairly good tri-nucleosome arrays. Even with Chd1, there is quite a bit of subnucleosomal product generated at fairly low MNase concentrations. The authors should present a longer MNase digestion series until the limit digest is observed*.

To test this idea, we performed a series of experiments with higher MNase concentrations. We found, however, that we were not able to obtain small chromatin fragments at the high concentrations of mutant Chd1 proteins used (e.g., the Chd1 might be binding to the DNA and inhibiting nucleases). Hence, in this specific case, we have carried out the experiments but were not able to obtain unambiguous data.

[Editors’ note: the authors requested clarification from the editors, which is shown below.]

*1) First, the manuscript (and title) seem to imply at times that CHD1 alone has chromatin assembly functions, hence comment 1. The authors should simply emphasize throughout the text that the CHD1 chromatin assembly function requires a histone chaperone and cannot be performed in vitro by CHD1 alone. With that information in mind, it will be pointless for them to repeat CHD1 chromatin assembly with mutant CHD1.*”

The revised manuscript now clearly indicates that chromatin assembly by Chd1 requires a histone chaperone and that Chd1 alone cannot assemble chromatin. The emphasis of this point will essentially eliminate any misunderstanding of the properties of Chd1 in the absence of a histone chaperone.

*2) Regarding the information garnered from 1D vs. 2D gels. Though 2D gels may provide more information, you are right in saying it is unlikely to change the conclusion of this manuscript. It may however help with the following comment.*”

Yes, we agree that it is unlikely that the addition of 2D gel data would change the conclusions of this manuscript. We further note that 2D gels are large and unwieldy. The use of 2D gels in this study would require the direct comparison of data from multiple independent gels, and it is questionable whether quantitative data from separate gels could be compared in a meaningful manner. For this study, we felt that it would be best to compare multiple 1D gel samples that are run in parallel on the same gel.

3) *Unfortunately, the reviewer is correct in stating that the main evidence suggesting chromatin assembly in mutant CHD1 solely relies on the DNA supercoiling analyses. Though technically challenging, a second technique unambiguously concluding the same therefore remains high desirable*.

We have successfully addressed this point. The editors indicated the desirability of demonstrating nucleosome assembly by the mutant Chd1 proteins by using a technique other than the DNA supercoiling assay.

To this end, we performed parallel chromatin assembly reactions with wild-type or mutant Chd1 proteins, digested the reaction products extensively with micrococcal nuclease, and then analyzed the resulting DNA fragments (new Figure 1) and chromatin particles (new Figure 1). As shown in the new Figure 1, we observed ATP-dependent stimulation of the formation of core particles by both wild-type and mutant Chd1 proteins. Hence, we now have used two different assays to provide evidence of nucleosome assembly by the mutant Chd1 proteins.